# Beyond Neural Collapse: Task-Intrinsic Geometry Governs Neural Representations in Modular Arithmetic

## Abstract

While Neural Collapse predicts that a $K$-class classifier should organize terminal representations as a $(K-1)$-dimensional simplex ETF, modular addition consistently enters a different regime: networks compress to a two-dimensional cyclic geometry in which both classifier weights and token embeddings lie on circles. We refine the explanation of this phenomenon in three directions. First, we formalize a layerwise non-uniform training mechanism: downstream classifier weights are driven by dense cross-entropy gradients into a rank-2 equiangular configuration before upstream embeddings fully reorganize, and once this classifier plane forms, backpropagated feature gradients constrain embedding motion to the same plane while weight decay suppresses orthogonal components. Second, after this subspace locking, the induced in-plane dynamics admit an entropy-regularized transport interpretation on $S^1$; combined with modular-addition labels, this reduces embedding formation to phase alignment, whose minimizers are single-frequency characters of $\mathbb{Z}/P\mathbb{Z}$ and hence equal-angle points on a circle. Third, we quantify why this solution prevails over Neural Collapse: a simplex ETF gains only an $O(1)$ advantage in cross-entropy, whereas the cyclic rank-2 solution enjoys a $\Theta(K)$ advantage under Schatten or weight-decay surrogates, yielding a critical threshold $\lambda_{\text{crit}} = \Theta(1/K)$. Our results explain both why classifier weights move first and why embeddings subsequently align with them, showing that grokking on modular arithmetic is governed not by maximal separation alone but by a task-structured trade-off between separation, symmetry, and complexity.

## 1 Introduction

Neural networks often appear to discover internal geometric codes that are far more structured than the supervision they receive. A major advance in making this precise is the Neural Collapse (NC) paradigm, which predicts that during terminal-phase training a $K$-class classifier should organize its class means and last-layer weights into a $(K-1)$-dimensional simplex equiangular tight frame (ETF), thereby maximizing mutual separation (Papyan et al., 2020; Lu & Steinerberger, 2022; Zhou et al., 2022). Subsequent work has refined this picture across depth, showing intermediate forms of collapse during feature learning and progressive layerwise compression rather than a purely last-layer phenomenon (Rangamani et al., 2023; Wang et al., 2025); see also the progressive feedforward-collapse perspective of Wang et al. (2024). Taken at face value, this line of theory suggests that more classes should generically induce more representational dimensions.

However, when we train neural networks on modular addition—computing $(a+b) \bmod P$ for inputs $a, b \in \{0, \ldots, P-1\}$ with $P$ prime—we observe the opposite tendency (see Fig. 1A). Instead of expanding toward the $(P-1)$ dimensions suggested by NC, the learned solution compresses into an essentially two-dimensional subspace. Both the learned embeddings and the classifier weights arrange themselves as points on circles, faithfully mirroring the cyclic structure of $\mathbb{Z}/P\mathbb{Z}$ (Fig. 1B,C). For $P = 97$, this is a 48-fold reduction in dimensionality, from the 96-dimensional simplex benchmark to a 2-dimensional cyclic code, achieved without any architectural prior that explicitly encodes group structure.

---

The authors used large language model (LLM) tools for language polishing, formatting assistance, and editorial revision. All scientific ideas, claims, proofs, experiments, and the final text were checked and approved by the authors.

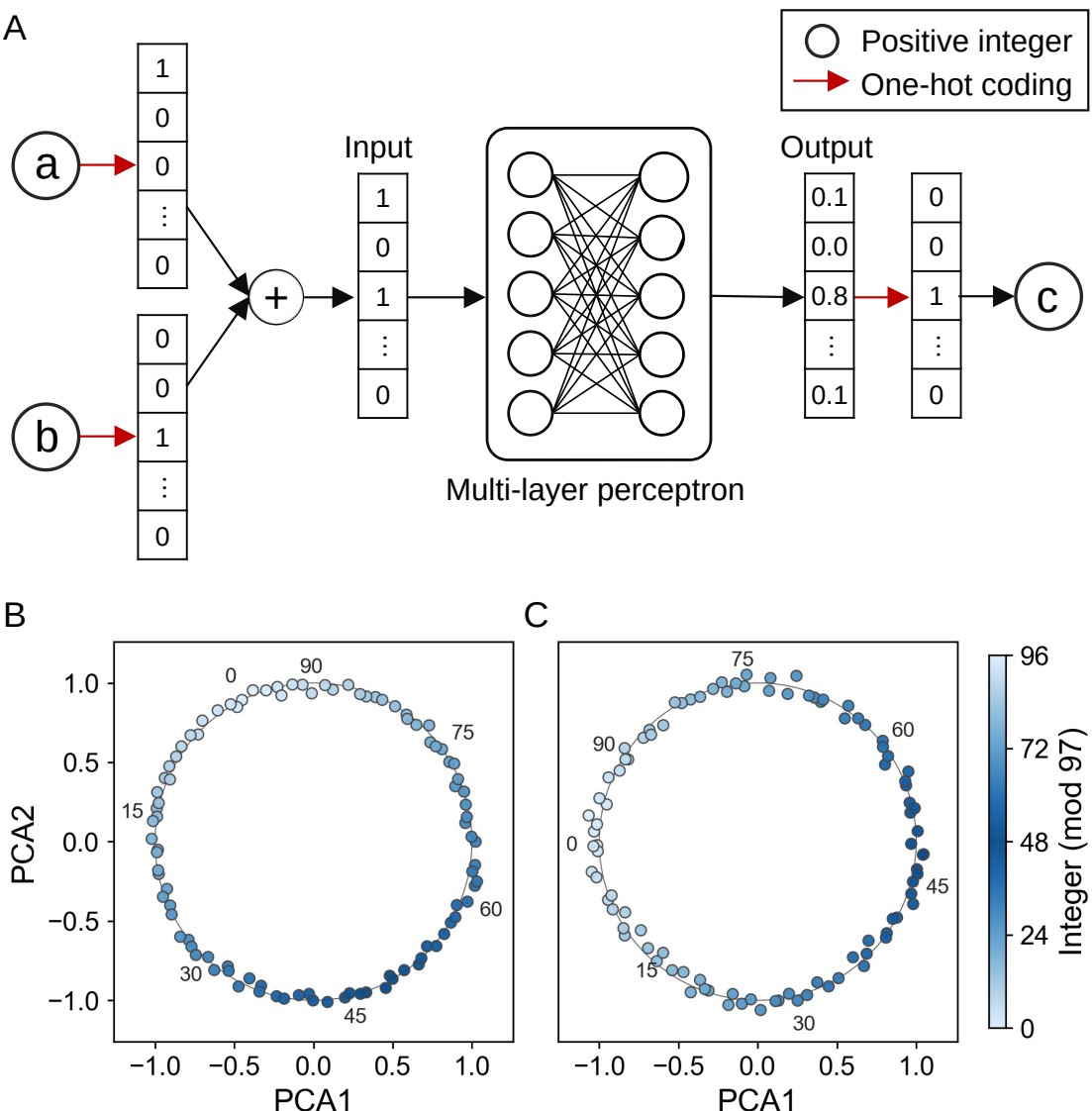

Figure 1: **Task setup and cyclic representations for modular addition modulo** 97**. (A)** Each input pair $(a, b)$ is one-hot encoded, concatenated, and processed by an MLP; the softmax output predicts $c = (a + b)$ mod 97. **(B)** PCA of the learned token embeddings after automorphism-based label reordering. **(C)** PCA of the output weights. The matched circular geometry in panels B and C shows that the network learns the task-intrinsic cyclic code rather than a generic high-dimensional separator.

This geometry is especially striking because modular addition also exhibits grokking: generalization remains poor for a long period and then rises abruptly once the structured solution is found (Power et al., 2022; Nanda et al., 2023; Liu et al., 2022; Gromov, 2023). Mechanistic and effective-theory studies already suggest that this transition is not homogeneous across layers. In particular, decoder/classifier learning and representation learning can proceed on different timescales (Liu et al., 2022); a recent two-clock formulation separates the fast decay of classification loss from slower representation simplification in grokking (Tan et al., 2026). In modular addition, weight decay can simplify the classifier and thereby induce the alignment forces that subsequently organize embeddings into circles or grids (Muşat, 2024). Recent embedding-centric analyses attribute delayed embedding reorganization to sparse, bilinear, token-dependent updates (AlquBoj et al.,

2025). These observations strongly suggest a missing theoretical ingredient up to now: the classifier geometry and the embedding geometry should not be explained as if they emerged simultaneously.

In this paper, we show that modular addition belongs to a distinct *task-structured low-rank regime*. The correct theoretical benchmark is not "maximal separation at any dimension," but rather "best separation compatible with the task symmetry under the regularization actually induced during training." Our main claim is that the network first organizes the last-layer weights into a rank-2 equiangular code, then pulls upstream embeddings into the same plane through backpropagated gradients, and finally phase-locks those embeddings into a cyclic character orbit. This perspective also clarifies how transport and geodesic language should enter the story: we do *not* need to claim that the full parameter trajectory is a Wasserstein geodesic; rather, after projection to the terminal classifier plane, the induced angular dynamics are naturally described by an entropy-regularized transport problem on the circle.

We make four contributions. **(1) Layerwise heterogeneity and classifier-first geometry.** We formalize a "classifier first, embeddings follow" mechanism: once the classifier collapses to a rank-2 plane, every task gradient that reaches the embeddings lies in that same plane, while orthogonal embedding components are damped by weight decay or implicit regularization. **(2) Geometric optimality of the circular codebook.** Within the two-dimensional terminal regime, we prove that the cross-entropy objective is uniquely minimized by an equiangular configuration of classifier directions. **(3) From alignment to characters.** Combining the additive feature model, an angular optimal-transport reformulation, and Fourier concentration, we show that embeddings aligned with the classifier must arrange as equal-angle points on a circle, i.e., as characters of the cyclic group. **(4) Why this is not Neural Collapse.** We derive an explicit trade-off showing that the simplex ETF gains only an $O(1)$ advantage in cross-entropy, whereas the cyclic rank-2 solution gains $\Theta(K)$ in regularization cost, yielding a sharp threshold $\lambda_{\mathrm{crit}} = \Theta(1/K)$ above which the cyclic solution is globally preferred.

Together, these results refine the status of NC. Modular addition is not a counterexample in which optimization somehow fails to reach the "right" high-dimensional configuration; it is a structured regime in which the optimal representation is genuinely low-rank because the task itself has a low-dimensional algebraic generator. The sudden emergence of the circle is therefore best understood as a hierarchical grokking event: first the classifier discovers the correct low-rank code, then the embeddings align to it, and only then does perfect generalization become possible.

## 2 Related Work

Neural Collapse has become a central framework for analyzing late-stage representation geometry in deep classification (Papyan et al., 2020). The phenomenon was first formalized through last-layer collapse, weight–feature alignment, and simplex ETF structure, and was then placed on firmer optimization-theoretic ground through unconstrained-feature and gradient-flow analyses (Lu & Steinerberger, 2022; Zhu et al., 2021; Zhou et al., 2022; Tirer & Bruna, 2022). More recent work extends this picture beyond the last layer: intermediate collapse has been observed during feature learning, later layers have been shown to progressively compress within-class variability while enhancing discrimination (Rangamani et al., 2023; Wang et al., 2025; Beaglehole et al., 2024), and a progressive feedforward-collapse viewpoint has been proposed for residual networks by Wang et al. (2024). These developments matter for our paper because they suggest that representational geometry is intrinsically layerwise rather than purely terminal. They also sharpen the comparison class in Section 6.1: once one departs from unconstrained-feature models and allows depth, factorization, or explicit regularization to bias the spectrum, low-rank alternatives to the simplex ETF become theoretically plausible rather than pathological (Jacot, 2023; Wang & Jacot, 2024; Zangrando et al., 2024; Súkeník et al., 2024).

At the same time, several works show that NC is not literally universal. Deviations have been reported in deeper or otherwise regularized models, and a growing theory literature argues that multi-layer regularization can favor lower-rank solutions than the simplex ETF predicted by shallow or unconstrained models (Dang et al., 2023; Tirer & Bruna, 2022; Jacot, 2023; Wang & Jacot, 2024; Zangrando et al., 2024; Súkeník et al., 2024). Our contribution is complementary to this literature: instead of studying a generic low-rank alternative, we identify the *specific* low-rank geometry selected by a structured task, namely the cyclic rank-2 code for modular addition.

On the grokking side, modular arithmetic has become a canonical setting for studying delayed generalization (Power et al., 2022). Effective-theory analyses explain grokking through competition between decoder fitting and representation learning (Liu et al., 2022); mechanistic studies show that hidden progress toward the eventual algorithm accumulates long before test accuracy jumps (Nanda et al., 2023; Gromov, 2023; Zhong et al., 2023); and recent task-specific work on modular addition gives increasingly explicit descriptions of the learned harmonic algorithm, the role of weight decay, and the lagging dynamics of the embedding layer (Mohamadi et al., 2024; Muşat, 2024; AlquBoj et al., 2025). Closely related to this layerwise view, Tan et al. (2026) formalize grokking through two training clocks, separating fast loss fitting from slower representation simplification using deep-linear theory and conditional ReLU reductions. We build directly on this line of work and focus on a gap that remains under-explained until now: why classifier weights appear to organize first, why embeddings subsequently align with them, and why the aligned solution is an equal-angle circle rather than a generic low-rank cloud.

Finally, our analysis uses optimal transport as a geometric reformulation of the angular cross-entropy objective. OT has become an influential language for structured representation learning, from Sinkhorn regularization to Wasserstein-based objectives and domain adaptation (Cuturi, 2013; Peyré & Cuturi, 2019; Arjovsky et al., 2017; Courty et al., 2017; Gai et al., 2024). In our setting, the relevant object is not a transport map in ambient Euclidean parameter space, but an induced transport problem on the circle once the terminal classifier plane has formed. This viewpoint connects modular-addition grokking to classical energy-minimization results on $S^1$, where regular polygons play a distinguished role under completely monotone potentials (Cohn & Kumar, 2007; Cohn et al., 2010).

## 3 Background and Preliminaries

### 3.1 Neural Collapse, Intermediate Collapse, and Equiangular Tight Frames

Neural Collapse (NC) characterizes the geometry of learned representations in the terminal phase of training. It manifests as four classical behaviors: (NC1) within-class variability vanishes as features collapse to their class means; (NC2) these class means form a simplex ETF in $\mathbb{R}^{K-1}$; (NC3) classifier weights align with the class means; and (NC4) the decision rule converges to nearest-class-mean classification (Papyan et al., 2020). In its cleanest form, this theory says that $K$ classes naturally induce a $(K-1)$-dimensional geometry when the dominant goal is separation.

Mathematically, a standard simplex ETF consists of $K$ vectors $\{\boldsymbol{v}_k\}_{k=1}^{K}$ in $\mathbb{R}^{K-1}$ such that for all $i, j$,

$$\boldsymbol{v}_i^\top \boldsymbol{v}_j = \begin{cases} 1, & i = j, \\ -\frac{1}{K-1}, & i \neq j. \end{cases}$$

This configuration maximizes mutual separation under a symmetric correlation criterion. However, recent depth-wise studies indicate that the representational path toward this configuration can be non-uniform across layers, and recent regularized deep-feature models show that high-rank NC geometries need not remain globally optimal once low-rank bias is taken seriously (Rangamani et al., 2023; Wang et al., 2025; Zangrando et al., 2024; Súkeník et al., 2024).

### 3.2 Modular Arithmetic and Cyclic Homomorphisms

We focus on the modular addition task defined on the cyclic group $G = \mathbb{Z}/P\mathbb{Z} = \{0, 1, \ldots, P-1\}$ for a prime $P$. The algebraic structure of $G$ is captured by its characters into the unit circle, namely maps $\chi_m : G \to S^1 \subset \mathbb{R}^2$ of the form

$$\chi_m(k) = \left[\cos\left(\frac{2\pi mk}{P}\right), \sin\left(\frac{2\pi mk}{P}\right)\right]^\top, \qquad m \in \{1, \ldots, P-1\}.$$

Geometrically, this places the group elements at equal angular intervals on a circle. When $\gcd(m, P) = 1$, the map is injective and preserves cyclic order up to permutation. A central question of this paper is why a neural network trained only on class labels should rediscover precisely this representation.

### 3.3 Grokking, Cross-Entropy, and Layerwise Implicit Bias

The modular addition task is known to exhibit grokking (Power et al., 2022): training accuracy reaches near perfection long before test accuracy does, and generalization emerges only after a long delay. Effective-theory analyses attribute this to a competition between fast decoder/classifier fitting and slower representation learning, with structured representations appearing only in the narrow regime where memorization and compression are properly balanced (Liu et al., 2022). A complementary two-clock perspective distinguishes fast fitting from slower representation simplification and gives a deep-linear theoretical account of this separation (Tan et al., 2026). In modular addition, recent training-dynamics studies further argue that weight decay prevents the classifier from overfitting pairwise sums, thereby creating the alignment force that eventually organizes embeddings into circles or grids (Muşat, 2024). Embedding-specific analyses sharpen this picture by showing that token embeddings receive sparse, frequency-dependent, and bilinearly coupled updates, which can cause them to lag behind downstream linear layers (AlquBoj et al., 2025).

We analyze this setting under the standard cross-entropy loss with temperature $\tau$:

$$\mathcal{L}_{\mathrm{CE}}(\boldsymbol{z}, y, \tau) = -\log \frac{\exp(z_y/\tau)}{\sum_{k=1}^{K} \exp(z_k/\tau)},$$

where $z_k = \boldsymbol{w}_k^\top \boldsymbol{h}$ is the logit for class $k$. Assuming terminal-phase NC-style alignment, where features and weights are aligned up to a scalar factor $s$ (so that $\boldsymbol{h} \propto \boldsymbol{w}_y$ implies $z_k \approx s\,\boldsymbol{w}_k^\top \boldsymbol{w}_y$), the effective inverse temperature becomes $\tilde{\tau}^{-1} = s/\tau$.

To understand dimensionality selection, we compare the cross-entropy term against the explicit or implicit low-rank bias induced during training. Following the literature on matrix factorization and deep linear networks, we use Schatten-$q$ norms $\|\boldsymbol{W}\|_{S_q} = (\sum_i \sigma_i^q)^{1/q}$ with $q < 2$ as a convenient surrogate: such penalties favor solutions with fewer non-zero singular values and hence lower effective rank (Arora et al., 2019; Gunasekar et al., 2017; Jacot, 2023; Wang & Jacot, 2024).

## 4 Problem Setup

We study neural networks learning modular addition as a $P$-class classification problem (Fig. 1A):

$$f : \{0, 1, \ldots, P-1\}^2 \to \{0, 1, \ldots, P-1\}, \qquad f(a, b) = (a + b) \bmod P.$$

Inputs are concatenated one-hot vectors in $\mathbb{R}^{2P}$, processed through a two-layer ReLU network with token embeddings $\{\mathbf{e}_i\}_{i=0}^{P-1}$ and last-layer weights $\{\mathbf{w}_k\}_{k=0}^{P-1}$.

A robust empirical regularity across modular-addition studies is that the learned solution spontaneously organizes into a two-dimensional circular structure (Liu et al., 2022; Gromov, 2023; Nanda et al., 2023; Zhong et al., 2023; Muşat, 2024; Mohamadi et al., 2024; AlquBoj et al., 2025):

$$\mathbf{e}_i \approx r_e \begin{pmatrix} \cos(2\pi i/P) \\ \sin(2\pi i/P) \end{pmatrix}, \qquad \mathbf{w}_k \approx r_w \begin{pmatrix} \cos(2\pi k/P) \\ \sin(2\pi k/P) \end{pmatrix}.$$

This sharply contradicts the NC prediction of $(P-1)$ dimensions while preserving the cyclic group structure. Equally important for our purposes, the temporal ordering in these studies is not perfectly synchronous: the last-layer geometry typically becomes visibly organized before the full embedding geometry stabilizes, which motivates the layerwise analysis developed below.

## 5 A Layered Theory of the Circular Solution

The empirical picture is not that all layers collapse at once. Rather, modular-addition training exhibits a sequence of geometric selections: the classifier first settles on a low-dimensional code, that code then constrains the admissible embedding updates, and only inside the resulting plane does the task symmetry pick the final circular arrangement. Presenting the theory in this order removes much of the apparent mystery around the embeddings.

We therefore separate three questions. First, what classifier geometry is preferred once the terminal decision space is effectively two-dimensional? Second, why do the last-layer weights appear to move earlier than the embeddings? Third, after the embedding dynamics have been locked into the classifier plane, why does alignment with the weights produce an equal-angle orbit rather than an arbitrary low-rank cloud?

### 5.1 Classifier Geometry in the Terminal Plane

Empirically, the trained last-layer weight matrix has effective rank 2. We take this terminal-plane regime as the starting point of the analysis. Once the logits are constrained to depend on two-dimensional directions, the optimization problem becomes purely angular, and the regular polygon emerges as the unique optimum.

**Theorem 5.1** (Loss–angular reduction). *Assume balanced classes and terminal-phase feature–classifier alignment with a common logit scale $s > 0$: $\mathbf{h}_{k,i} = s\,\mathbf{w}_k$, where each $\mathbf{w}_k \in \mathbb{R}^2$ is a unit vector. Let $\tilde{\tau} = \tau/s$ be the effective temperature. Then minimizing the sample-averaged cross-entropy is equivalent, up to an additive constant, to minimizing*

$$f(\theta_1, \ldots, \theta_K) = \sum_{k=1}^{K} \log\Big(1 + \sum_{j \neq k} \exp\Big(\frac{\cos(\theta_k - \theta_j) - 1}{\tilde{\tau}}\Big)\Big), \tag{5.1}$$

*where $\theta_k$ is the polar angle of $\mathbf{w}_k$.*

*Proof idea.* Under the alignment assumption, logits depend only on pairwise inner products $\mathbf{w}_k^\top \mathbf{w}_j = \cos(\theta_k - \theta_j)$; the scale $s$ is absorbed into the effective temperature $\tilde{\tau}$. The reduction is written out in Appendix A.1.

**Theorem 5.2** (Equiangular optimality in the classifier plane). *The potential $f$ in (5.1) achieves its unique global minimum, up to a common rotation, at the equiangular configuration*

$$\theta_k^* = \theta_0 + \frac{2\pi k}{K}, \qquad k = 0, 1, \ldots, K-1,$$

*for an arbitrary $\theta_0 \in \mathbb{R}$.*

*Proof idea.* The kernel $h(\theta) = \exp((\cos\theta - 1)/\tilde{\tau})$ is a completely monotone function of the squared chordal distance on $S^1$. Classical circle-energy results therefore identify the regular $K$-gon as the unique minimizer of the associated pairwise interaction, and a sandwich argument transfers that optimality to the log-sum-exp potential $f$; see Appendix A.2 and Cohn & Kumar (2007); Cohn et al. (2010).

**Theorem 5.3** (Terminal-phase weight–feature alignment). *Consider the last-layer objective with weights $W = [\mathbf{w}_1, \ldots, \mathbf{w}_K] \in \mathbb{R}^{2 \times K}$, balanced class means $\{\boldsymbol{\mu}_k\} \subset \mathbb{R}^2$, effective temperature $\tilde{\tau} > 0$, and weight decay $\lambda > 0$:*

$$\min_W \frac{1}{K} \sum_{k=1}^{K} \Big[-\frac{1}{\tilde{\tau}} \mathbf{w}_k^\top \boldsymbol{\mu}_k + \log \sum_{j=1}^{K} e^{\mathbf{w}_j^\top \boldsymbol{\mu}_k / \tilde{\tau}}\Big] + \lambda \|W\|_F^2. \tag{5.2}$$

*At any stationary point in the terminal phase, $\mathbf{w}_k \parallel \boldsymbol{\mu}_k$ for every $k$. If $\mathrm{span}\{\boldsymbol{\mu}_k\}$ is two-dimensional, then the induced angular objective has the unique minimizer (up to rotation) in which both $\{\mathbf{w}_k\}$ and $\{\boldsymbol{\mu}_k\}$ form regular $K$-gons with class-wise alignment.*

*Proof idea.* First-order optimality forces each $\mathbf{w}_k$ to lie in the span of the class means. In the balanced setting, symmetry and correct classification then imply $\mathbf{w}_k \parallel \boldsymbol{\mu}_k$, after which Theorem 5.2 applies directly. Appendix A.3 gives the detailed argument.

### 5.2 Layerwise Heterogeneity: Why the Classifier Moves First

The three theorems above characterize the *terminal* classifier geometry, but they do not yet explain the ordering observed in training. Existing grokking analyses already point to a non-uniform dynamics: decoder fitting is faster than representation learning (Liu et al., 2022); in modular addition, weight decay simplifies the classifier and thereby creates the alignment force that later organizes the embeddings (Muşat, 2024);

and recent embedding-centric analyses explain delayed embedding reorganization through sparse, bilinear, token-dependent updates (AlquBoj et al., 2025). The next proposition identifies the structural reason behind these observations.

**Proposition 5.4** (Classifier-induced subspace locking)**.** *Let $\ell_{ij}$ denote the cross-entropy loss for sample $(i,j)$ with label $y_{ij}$, logits $z_{ij} = W^\top \mathbf{h}_{ij} \in \mathbb{R}^K$, and probabilities $p_{ij} = \mathrm{softmax}(z_{ij}/\tau)$. Then*

$$\nabla_{\mathbf{h}_{ij}} \ell_{ij} = \frac{1}{\tau} W\big( p_{ij} - \boldsymbol{\delta}_{y_{ij}} \big) \in \mathrm{span}(W), \tag{5.3}$$

*where $\boldsymbol{\delta}_{y_{ij}}$ is the standard basis vector of the correct class. Under the additive feature model (5.5), $\mathbf{h}_{ij} \approx A(\mathbf{e}_i + \mathbf{e}_j)$, and hence*

$$\nabla_{\mathbf{e}_i} \ell_{ij} \approx A^\top \nabla_{\mathbf{h}_{ij}} \ell_{ij}, \qquad \nabla_{\mathbf{e}_j} \ell_{ij} \approx A^\top \nabla_{\mathbf{h}_{ij}} \ell_{ij} \in A^\top \, \mathrm{span}(W). \tag{5.4}$$

*Consequently, once the classifier matrix $W$ has effective rank 2, all task-relevant embedding updates are confined to the same two-dimensional subspace $A^\top \, \mathrm{span}(W)$.*

*Proof idea.* The gradient formula in (5.3) is standard for softmax cross-entropy, and (5.4) follows by the chain rule through the local additive map $\mathbf{h}_{ij} \approx A(\mathbf{e}_i + \mathbf{e}_j)$. Appendix A.4 also shows how explicit weight decay removes embedding components orthogonal to this task-supported plane.

This is the sense in which modular-addition grokking is *hierarchical.* The classifier and embeddings remain coupled throughout training, but they are not driven with equal strength. Dense gradients on the last layer can organize the classifier early, whereas the embeddings receive weaker, token-dependent signals filtered through the current classifier. Once the classifier plane forms, the embedding dynamics no longer explore the ambient space freely; they relax inside a plane that has already been selected downstream.

## 5.3 From Plane Locking to Circular Embeddings

What remains is to explain the in-plane geometry. Subspace locking alone only says that the embeddings end up in the classifier plane; it does *not* yet say why they land on a circle, nor why the angular positions are equally spaced. Mechanistic analyses of modular addition already suggest that successful solutions are organized by a single dominant harmonic or phase code (Gromov, 2023; Zhong et al., 2023; Mohamadi et al., 2024). Our goal here is to recover that picture from the post-collapse optimization itself. After locking, the residual optimization problem is predominantly angular, and modular addition converts that angular optimization into a phase-matching problem on the cyclic group.

### 5.3.1 Additive Feature Model and Half-Angle Geometry

**Working model.** We model the terminal two-dimensional regime by the additive approximation

$$\mathbf{h}_{ij} \approx A(\mathbf{e}_i + \mathbf{e}_j) \in \mathbb{R}^2, \tag{5.5}$$

where $A \in \mathbb{R}^{2 \times d}$ is the effective projection into the classifier plane and $\mathbf{e}_i \in \mathbb{R}^d$ is the token embedding for symbol $i$. Writing $\mathbf{v}_i := A\mathbf{e}_i$, we have $\mathbf{h}_{ij} \approx \mathbf{v}_i + \mathbf{v}_j$. This is not an architectural axiom; it is a terminal-phase description of the regime reached in our experiments, and it isolates the degrees of freedom that still matter once the classifier span has collapsed to two dimensions. In particular, it matches harmonic descriptions of the learned modular-addition circuit, where token-level phases are composed and then decoded by a downstream linear readout (Gromov, 2023; Zhong et al., 2023; Mohamadi et al., 2024).

**Half-angle decomposition.** Let $\mathbf{v}_i$ and $\mathbf{v}_j$ be nonzero vectors with polar angles $\theta_i, \theta_j$ and magnitudes $\rho_i, \rho_j$. Defining $\Delta_{ij} = \theta_j - \theta_i$ and

$$\delta_{ij} := \mathrm{atan2}\Big( (\rho_j - \rho_i) \sin\big(\tfrac{\Delta_{ij}}{2}\big), \ (\rho_i + \rho_j) \cos\big(\tfrac{\Delta_{ij}}{2}\big) \Big),$$

we obtain

$$\alpha_{ij} = \frac{\theta_i + \theta_j}{2} + \delta_{ij} \pmod{2\pi}, \tag{5.6}$$

where $\alpha_{ij}$ is the polar angle of $\mathbf{v}_i + \mathbf{v}_j$. In particular, if $\rho_i = \rho_j$, then $\delta_{ij} = 0$ and $\alpha_{ij}$ is exactly the angular bisector $(\theta_i + \theta_j)/2$. Appendix B.1 gives the derivation.

This decomposition explains why the terminal embedding problem becomes angular. Once the task-relevant motion is trapped in a two-dimensional plane, only radii and phases remain. Empirically, balanced sampling and regularization tend to homogenize the radii, while the label $y_{ij} = (i + j) \bmod P$ constrains the combined phase. In the homogeneous regime, the feature angle is therefore controlled by the half-angle combination $(\theta_i + \theta_j)/2$.

### 5.3.2 Angular Reduction of the Post-Collapse Loss

Fix the equiangular classifier angles $\varphi_k = \varphi_0 + 2\pi k/P$ and write the sample feature as $\mathbf{h}_{ij} = r_{ij}(\cos\alpha_{ij}, \sin\alpha_{ij})$. Let $\beta_{ij} = r_{ij}/\tau$ and define the circular cost

$$c(\alpha, \varphi) := 1 - \cos(\alpha - \varphi). \tag{5.7}$$

The next proposition packages the post-collapse loss into a single statement: the leading term is precisely angular matching to the class prototype, and the only correction is a rapidly decaying $P$-periodic remainder.

**Proposition 5.5** (Angular reduction of post-collapse cross-entropy). *For each sample $(i, j)$ with label $y_{ij} = (i + j) \bmod P$,*

$$\mathcal{L}_{ij} = C(\beta_{ij}) + \beta_{ij}\, c(\alpha_{ij}, \varphi_{y_{ij}}) + R_P(\beta_{ij}, \alpha_{ij}), \tag{5.8}$$

*where*

$$C(\beta) := \log(PI_0(\beta)) - \beta,$$

*and*

$$R_P(\beta, \alpha) := \log\left(1 + 2\sum_{\ell=1}^{\infty} \frac{I_{\ell P}(\beta)}{I_0(\beta)} \cos(\ell P(\alpha - \varphi_0))\right).$$

*Define*

$$\eta_P(\beta) := 2\sum_{\ell=1}^{\infty} \frac{I_{\ell P}(\beta)}{I_0(\beta)}.$$

*Then*

$$\left| 2\sum_{\ell=1}^{\infty} \frac{I_{\ell P}(\beta)}{I_0(\beta)} \cos(\ell P(\alpha - \varphi_0)) \right| \leq \eta_P(\beta),$$

*and, whenever $\eta_P(\beta) < 1$,*

$$\left| R_P(\beta, \alpha) \right| \leq \frac{\eta_P(\beta)}{1 - \eta_P(\beta)}. \tag{5.9}$$

*Moreover, using $I_n(\beta) \leq (\beta/2)^n/n!$ for $n \geq 0$ gives*

$$\eta_P(\beta) \leq 2\sum_{\ell=1}^{\infty} \frac{(\beta/2)^{\ell P}}{(\ell P)!}.$$

*In particular, if $\eta_P(\beta) \leq \frac{1}{2}$, then $\left| R_P(\beta, \alpha) \right| \leq 2\eta_P(\beta)$.*

*Proof idea.* Two standard ingredients are combined in Appendix B.2. First, Fenchel–Young duality identifies the softmax distribution with the optimizer of an entropy-regularized assignment problem on the prototype angles, so cross-entropy becomes an angular transport objective on $S^1$. Second, summing the partition function over the regular $P$-gon and using the Fourier–Bessel expansion isolates the class-dependent phase-alignment term and pushes every other contribution into a $P$-periodic remainder. The revised bound (5.9) makes explicit the small-tail condition under which this remainder is perturbative.

The transport language is therefore useful, but only at the right level. We do *not* claim that the full parameter trajectory of the network is globally a Wasserstein geodesic in parameter space. The relevant statement is local to the terminal plane: once features are projected to span$(W)$, each sample angle is softly assigned to the equiangular prototype angles under the circular cost $c(\alpha, \varphi)$, which is a smooth surrogate of squared geodesic distance on $S^1$ near alignment.

### 5.3.3 Character Selection via Fourier Concentration

Proposition 5.5 explains why alignment with the classifier is the right post-collapse objective, but it still leaves open which in-plane embedding configuration minimizes that objective. Under norm homogenization, the answer is rigid at leading order: the only minimizers of the dominant angular functional are cyclic characters.

**Theorem 5.6** (Character-orbit minimizers of the leading-order angular objective)**.** *Let $P$ be an odd prime. Under the additive feature model* (5.5)*, suppose the effective radii homogenize so that the correction term in* (5.6) *vanishes, i.e. $\delta_{ij} = 0$, and suppose $\beta_{ij}$ concentrates to a common value $\beta$. Consider the leading-order objective*

$$\mathcal{E}(\theta) := \sum_{i,j} \Big[ 1 - \cos\Big( \frac{\theta_i + \theta_j}{2} - \varphi_{(i+j) \bmod P} \Big) \Big]. \tag{5.10}$$

*Then every minimizer of $\mathcal{E}$ satisfies*

$$e^{i\theta_i/2} = c\,\omega^{mi}, \qquad \omega = e^{2\pi i/P}, \tag{5.11}$$

*for some $|c| = 1$ and $m \in (\mathbb{Z}/P\mathbb{Z})^\times$. Equivalently,*

$$\theta_i = \theta_0 + \frac{4\pi m}{P} i \pmod{2\pi}, \tag{5.12}$$

*up to a global rotation and the automorphisms of $\mathbb{Z}/P\mathbb{Z}$.*

*Proof idea.* When the correction term in (5.6) vanishes, the dominant loss becomes a phase-matching objective in the half-angles $\theta_i/2$. Writing $z_i = e^{i\theta_i/2}$ turns the total energy into a convolutional inner product against the target code $u_t = e^{i\varphi_t}$. The discrete Fourier transform diagonalizes this convolution; only one Fourier line can interact with the equiangular codebook, and Parseval forces the optimum to place all spectral mass on that line. Appendix B.3 spells out the reduction and the resulting character form.

**Remark 5.7** (From leading-order alignment to the exact loss)**.** *Theorem 5.6 is stated for the leading-order angular objective* (5.10)*, which is the level at which the Fourier argument is exact. Proposition 5.5 shows that the exact post-collapse cross-entropy differs from this objective by a $P$-periodic remainder whose size is controlled by* (5.9)*. Thus, whenever the tail parameter $\eta_P(\beta)$ is small, the exact objective is a perturbation of the character-selecting angular functional.*

The point is stronger than mere plane alignment. The classifier plane does not just tell the embeddings *where* to live; combined with the modular-addition labels, it tells them *how* to arrange themselves inside that plane. This conclusion is consistent with mechanistic analyses in which the learned algorithm collapses onto a single dominant Fourier mode (Gromov, 2023; Zhong et al., 2023; Mohamadi et al., 2024). The circular code is therefore not an aesthetic afterthought but the natural harmonic orbit selected by the task symmetry.

## 6 Trade-off Analysis: Cyclic Rank-2 Geometry versus the Simplex ETF

Sections 5.1–5.3.3 explain why modular addition selects a regular $P$-gon once the terminal dynamics have entered a two-dimensional plane. To complete the story, we must compare that structured rank-2 solution with the canonical NC competitor, namely the simplex ETF. The point of this section is not to deny the NC theory developed for unconstrained-feature or separation-dominated models (Papyan et al., 2020; Zhu et al., 2021; Zhou et al., 2022; Dang et al., 2023), but to show that modular addition lives in a different comparison class. Here the labels are linked by cyclic translations, and the training dynamics carry an explicit or implicit low-rank bias. In such a regime, the relevant question is not which geometry maximizes pairwise separation in *some* ambient dimension, but which geometry minimizes the *regularized* objective among task-compatible solutions (Jacot, 2023; Wang & Jacot, 2024; Zangrando et al., 2024; Súkeník et al., 2024; Beaglehole et al., 2024).

### 6.1 Why the optimum is not a simplex ETF

Neural Collapse is most compelling when class means are effectively free variables and the dominant objective is mutual separation. This is exactly the comparison class captured by unconstrained-feature and deep-linear

analyses: when the task provides $K$ unrelated classes and regularization does not strongly penalize rank, the simplex ETF is the natural maximally separated optimum (Zhu et al., 2021; Zhou et al., 2022; Dang et al., 2023). Modular addition violates both premises. First, class identities are not unrelated: class $k + 1$ is a phase shift of class $k$, so the label space already carries a one-parameter cyclic structure. Second, the learning dynamics of modular addition are not rank-neutral: weight decay, factorization, and depth all bias the network toward simpler spectral organizations (Muşat, 2024; AlquBoj et al., 2025; Jacot, 2023; Wang & Jacot, 2024; Súkeník et al., 2024).

This is why the relevant comparison is not "simplex versus failure to separate." Reverse-engineering studies of modular addition repeatedly find harmonic or circulant algorithms rather than generic simplex packings (Gromov, 2023; Zhong et al., 2023; Mohamadi et al., 2024). Our claim is therefore not that NC is "wrong"; rather, modular addition lies in a structured regime in which the cyclic rank-2 code is almost as discriminative as the ETF but dramatically cheaper to realize. The trade-off analysis below makes that statement quantitative.

**Remark 6.1** (Task structure changes the comparison class). *The simplex ETF remains the canonical optimum for unconstrained, separation-dominated models. Our result concerns a* regularized structured *problem: among representations that must respect modular addition and are penalized for rank or norm complexity, the cyclic rank-2 code attains a better total objective. Recent low-rank-bias results explain why such alternatives can be favored in deep networks; the modular-addition analysis identifies the particular alternative selected by the task symmetry.*

## 6.2 Comparative Analysis of Geometric Configurations

We compare two candidate geometries under the regularized objective

$$\mathcal{L}_{\text{tot}}(W) \coloneqq \mathcal{L}_{\text{CE}}(W) + \lambda \mathcal{R}(W), \tag{6.1}$$

where $\mathcal{R}(W)$ is either the Schatten surrogate $\|W\|_{S_q}^q$ with $q < 2$, or, in the factorized weight-decay proxy discussed below, the nuclear norm $\|W\|_*$. The candidates are:

1. **Cyclic rank-2 configuration** ($W_{\text{cyc}}$): unit columns $\mathbf{w}_k \in \mathbb{R}^2$ placed at angles $2\pi k/K$ on the circle.

2. **Simplex ETF configuration** ($W_{\text{ETF}}$): unit columns $\mathbf{w}_k \in \mathbb{R}^{K-1}$ satisfying $\mathbf{w}_i^\top \mathbf{w}_j = -1/(K-1)$ for $i \neq j$.

### 6.2.1 Cross-Entropy: The Separation Gap

First we quantify the classification price of compressing $K$ classes into two dimensions.

**Theorem 6.2** (Asymptotic loss gap). *For $K \to \infty$ and effective temperature $\tilde{\tau} > 0$, the gap in cross-entropy loss between the cyclic rank-2 configuration and the simplex ETF converges to a constant depending only on $\tilde{\tau}$:*

$$\Delta_{\text{CE}} \coloneqq \mathcal{L}_{\text{CE}}(W_{\text{cyc}}) - \mathcal{L}_{\text{CE}}(W_{\text{ETF}}) = \log I_0(1/\tilde{\tau}) + o(1),$$

*where $I_0$ is the modified Bessel function of the first kind.*

*Proof idea.* The ETF has a single off-diagonal correlation level $-1/(K-1)$, whereas the cyclic code samples the full cosine profile on the circle. The corresponding log-sum-exp term is therefore a Riemann sum over the circle, and its large-$K$ limit is governed by the defining integral of $I_0$. Appendix C.1 contains the detailed asymptotic calculation.

**Remark 6.3** (Bounded disadvantage). *Theorem 6.2 makes the first part of the comparison precise: the classification disadvantage of the cyclic code is only $O(1)$, not $O(K)$. In other words, the ETF improves the packing constant, but it does not buy an extensive gain in the supervised loss. This bounded gap is the sense in which the cyclic code remains a serious competitor to the NC geometry.*

### 6.2.2 Regularization: The Complexity Gap

Next we analyze the complexity term. For $q < 2$, Schatten penalties favor concentrating spectral mass on fewer singular directions, which is exactly the regime emphasized by recent low-rank-bias results for deep and factorized models (Jacot, 2023; Wang & Jacot, 2024; Súkeník et al., 2024).

**Theorem 6.4** (Schatten scaling gap). *For any $q < 2$ and $K \geq 4$,*

$$\|W_{\mathrm{cyc}}\|_{S_q}^q = 2\left(\frac{K}{2}\right)^{q/2}, \qquad \|W_{\mathrm{ETF}}\|_{S_q}^q = (K-1)\left(\frac{K}{K-1}\right)^{q/2}.$$

*Consequently,*

$$\Delta_q := \|W_{\mathrm{cyc}}\|_{S_q}^q - \|W_{\mathrm{ETF}}\|_{S_q}^q = -\Theta(K).$$

*Proof idea.* For the cyclic code, $W_{\mathrm{cyc}}W_{\mathrm{cyc}}^\top = (K/2)I_2$, so only two singular values are nonzero. For the ETF, the Gram matrix has $K-1$ identical nonzero eigenvalues. When $q < 2$, the penalty is minimized by concentrating spectral mass rather than spreading it across many directions, which gives the cyclic code a linear advantage in $K$. Appendix C.2 gives the full derivation.

**Proposition 6.5** (Connection to explicit factorized weight decay). *Assume the effective classifier matrix is realized as $W = UV^\top$ and training uses layerwise Frobenius decay $\frac{\lambda}{2}(\|U\|_F^2 + \|V\|_F^2)$. Then eliminating the factorization induces the variational penalty*

$$\inf_{W=UV^\top} \frac{1}{2}\left(\|U\|_F^2 + \|V\|_F^2\right) = \|W\|_*.$$

*Therefore explicit factorized weight decay corresponds to the nuclear norm, i.e. the case $q = 1$. For the two candidate geometries,*

$$\|W_{\mathrm{cyc}}\|_* = \sqrt{2K}, \qquad \|W_{\mathrm{ETF}}\|_* = \sqrt{K(K-1)},$$

*and hence the cyclic code again enjoys a $\Theta(K)$ regularization advantage.*

*Proof idea.* The variational characterization of the nuclear norm is obtained by taking an SVD of $W$ and balancing the square roots of the singular values across the two factors. Appending the exact singular-value formulae from Theorem 6.4 then gives the stated expressions. Appendix C.3 records the details.

### 6.2.3 Phase Transition in the Combined Objective

Combining the bounded cross-entropy loss gap with the linear regularization gain reveals a sharp threshold.

**Theorem 6.6** (Global optimality threshold). *For $K$ sufficiently large, the cyclic rank-2 configuration achieves a lower regularized objective than the simplex ETF whenever the regularizer is either $\mathcal{R}(W) = \|W\|_{S_q}^q$ for some fixed $q < 2$ or $\mathcal{R}(W) = \|W\|_*$ via Proposition 6.5, and the regularization strength satisfies*

$$\lambda_{\mathrm{crit}} \asymp \frac{\Delta_{\mathrm{CE}}}{|\Delta_q|} = \Theta\left(\frac{1}{K}\right).$$

*In particular, for the weight-decay proxy $q = 1$, the same $\Theta(1/K)$ threshold holds.*

*Proof idea.* The cyclic code wins exactly when its bounded cross-entropy disadvantage is outweighed by its linear regularization advantage. The threshold is therefore obtained by comparing Theorem 6.2 with either Theorem 6.4 or Proposition 6.5; Appendix C.4 writes the objective difference explicitly and verifies the claimed $\Theta(1/K)$ scaling in both cases.

**Example 6.7** (Numerical validation for $K = 97$). *Take the experimentally relevant setting $K = 97$, $q = 1$, and $\tilde{\tau} = 1$. Since $q = 1$ coincides with the nuclear-norm surrogate induced by factorized weight decay, Proposition 6.5 gives*

$$\Delta_{\mathrm{CE}} \approx \log I_0(1) \approx \log(1.2661) \approx 0.236,$$

$$\|W_{\mathrm{cyc}}\|_* = \sqrt{2K} = \sqrt{194} \approx 13.93,$$

$$\|W_{\mathrm{ETF}}\|_* = \sqrt{K(K-1)} = \sqrt{97 \cdot 96} \approx 96.50,$$

$$|\Delta_1| \approx 82.57, \qquad \lambda_{\mathrm{crit}} \approx \frac{0.236}{82.57} \approx 2.86 \times 10^{-3}.$$

*Thus, for $K = 97$, a factorized weight-decay coefficient on the order of $10^{-3}$ is already sufficient for the regularization gain of the cyclic rank-2 code to dominate its bounded cross-entropy disadvantage. The calculation also makes the scaling transparent: the cross-entropy term is essentially constant in $K$, whereas the nuclear-norm saving grows linearly with the number of classes.*

### 6.3 Implications: Regularization as Structure Selection

The comparison above sharpens the relationship between our theory and Neural Collapse. NC predicts the geometry preferred by separation-dominated objectives; our analysis identifies the geometry preferred when *the task itself* already supplies a low-dimensional harmonic parameterization and regularization penalizes unused directions. This viewpoint is consistent with recent work arguing that deep networks can deviate from the simplex ETF once low-rank bias becomes competitive with maximal separation (Súkeník et al., 2024; Zangrando et al., 2024; Beaglehole et al., 2024). The conclusion is not that NC breaks down arbitrarily, but that the comparison class changes once one moves from generic classification to structured algorithmic tasks.

The trade-off has a particularly transparent form in modular addition:

- **Cross-entropy cost:** replacing the ETF by the cyclic code incurs only a bounded $O(1)$ penalty, quantified by $\log I_0(1/\tilde{\tau})$.

- **Complexity gain:** the same replacement saves $\Theta(K)$ in either Schatten or factorized-weight-decay regularization.

The structured solution is therefore favored as soon as the effective regularization exceeds an order-$1/K$ threshold. This is exactly the regime in which recent low-rank-bias theories predict departures from deep NC (Súkeník et al., 2024; Zangrando et al., 2024; Beaglehole et al., 2024). What our analysis adds is the *identity* of the winning low-rank solution: modular addition does not merely prefer "some" compressed representation, but the specific cyclic code compatible with the group law.

Viewed this way, the two-dimensional circle is not an optimization accident and not a cosmetic variant of NC. It is the structured optimum of a task-aware trade-off between separation, symmetry, and complexity. That is why the classifier can settle into a rank-2 code before the embeddings fully reorganize, and also why the embeddings that eventually align with it are arranged on a regular polygon rather than on a simplex.

## 7 Conclusion and Discussion

This study uncovers a layered geometric mechanism governing how neural networks learn modular addition. We show that the standard $(K-1)$-dimensional ETF predicted by Neural Collapse is superseded by a task-structured rank-2 regime in which both classifier weights and embeddings lie on circles. More importantly, we refine the theory of *how* this happens. The classifier weights do not merely reflect an already organized embedding space; they organize first. Once the last layer contracts to a two-dimensional equiangular code, the backpropagated feature gradients force all task-relevant embedding motion into the same plane, after which the residual optimization problem becomes purely angular.

Within that plane, the cross-entropy loss admits a clean transport-based interpretation on $S^1$. Combined with modular-addition labels, this turns embedding learning into a phase-alignment problem whose minimizers are single-frequency characters of $\mathbb{Z}/P\mathbb{Z}$. This explains why alignment to the classifier does not produce an arbitrary low-rank cloud, but specifically an equal-angle circle. Finally, by comparing the cyclic rank-2 solution with the simplex ETF, we show that the ETF gains only an $O(1)$ advantage in cross-entropy, whereas the cyclic solution gains a decisive $\Theta(K)$ advantage in regularization cost. This yields a critical threshold $\lambda_{\mathrm{crit}} = \Theta(1/K)$ above which the structured 2D solution is globally preferred.

These findings substantially refine the interpretation of Neural Collapse. Our results do not say that NC is false; they say that NC describes a separation-dominated regime, while modular addition lies in a *symmetry-dominated low-rank regime*. In such problems, the network does not default to maximal separation in high dimensions. Instead, it discovers a minimal sufficient representation that encodes the generator of the task. From this viewpoint, grokking is a hierarchical event: first the classifier discovers the correct low-rank code, then the embeddings align with it, and only then does abrupt generalization occur.

**Broader implications.** Our results suggest three broader principles for representation learning:

- **Structure over separation.** When the gain from higher-dimensional separation is bounded but the complexity saving of a low-rank code grows with $K$, structured low-rank solutions can dominate the training objective.

- **Hierarchical collapse across layers.** In grokking-style algorithmic tasks, geometric organization need not arise uniformly across layers. Downstream classifiers can set the relevant low-dimensional subspace first, with upstream embeddings subsequently relaxing into it.

- **Geometry as algorithm.** The circular representation of $\mathbb{Z}/P\mathbb{Z}$ indicates that networks implement algorithms by physicalizing algebraic structure into geometry, rather than by memorizing isolated labels.

**Limitations and outlook.** Our theory still relies on balanced classes, near-terminal feature–weight alignment, and prime moduli. Extending the analysis to composite moduli, product groups, or non-Abelian settings may reveal tori or other structured manifolds in place of the circle. A second limitation is interpretive: we provide a geometric account of the induced angular dynamics after projection to the classifier plane, but we do not claim that the full parameter trajectory of the network is globally geodesic in parameter space. Finally, while we quantify the preference for the structured optimum, a sharper dynamical theory of when optimizers and schedules enter the classifier-first regime remains an important problem for future work.

In summary, deep networks trained on modular arithmetic do not simply expand dimensionality to separate classes. Guided by the trade-off between cross-entropy and regularization, they first identify a low-rank classifier code and then align the embedding layer with the underlying cyclic symmetry of the task. This viewpoint not only explains the striking 2D circles observed in grokking, but also offers a more general lesson: the geometry learned by a network is jointly determined by class separation, inter-layer dynamics, and the latent algebraic structure of the data.

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

## A    Proofs for the Classifier Geometry

### A.1   Proof of Theorem 5.1

*Proof.* For a sample from class $k$, the cross-entropy loss with temperature $\tau$ is

$$\mathcal{L}_{k,i} = \log\left(\sum_{j=1}^{K} \exp\left(\frac{\mathbf{w}_j^\top \mathbf{h}_{k,i}}{\tau}\right)\right) - \frac{\mathbf{w}_k^\top \mathbf{h}_{k,i}}{\tau}.$$

Under terminal-phase alignment, $\mathbf{h}_{k,i} = s\,\mathbf{w}_k$ with $\|\mathbf{w}_k\|_2 = 1$. Hence

$$\mathbf{w}_j^\top \mathbf{h}_{k,i} = s\,\mathbf{w}_j^\top \mathbf{w}_k = s\cos(\theta_k - \theta_j), \qquad \mathbf{w}_k^\top \mathbf{h}_{k,i} = s.$$

Substituting into the loss and dividing numerator and denominator by $e^{s/\tau}$ gives

$$\mathcal{L}_{k,i} = \log\Big(1 + \sum_{j\neq k} \exp\Big(\frac{s(\cos(\theta_k - \theta_j) - 1)}{\tau}\Big)\Big) = \log\Big(1 + \sum_{j\neq k} \exp\Big(\frac{\cos(\theta_k - \theta_j) - 1}{\tilde{\tau}}\Big)\Big),$$

where $\tilde{\tau} = \tau/s$. Averaging over balanced classes only introduces a constant multiplicity factor, so minimizing the sample-averaged cross-entropy is equivalent to minimizing (5.1). □

### A.2   Proof of Theorem 5.2

We prove the result in three steps: first compare the log-sum-exp objective with a pairwise energy, then minimize that pairwise energy using classical circle-energy arguments, and finally transfer the conclusion back to the original objective.

*Proof.* Fix $K \geq 2$ and $\tilde{\tau} > 0$. Let

$$h(t) := \exp\left(\frac{\cos t - 1}{\tilde{\tau}}\right), \qquad S_k(\theta) := \sum_{j\neq k} h(\theta_k - \theta_j), \qquad S_{\text{tot}}(\theta) := \sum_{k=1}^{K} S_k(\theta).$$

Then

$$f(\theta) = \sum_{k=1}^{K} \log\big(1 + S_k(\theta)\big).$$

**Step 1: a sandwich inequality.** Because $0 \leq S_k(\theta) \leq K - 1$, the derivative of $\psi(x) = \log(1 + x)$ satisfies $\psi'(x) \in [1/K, 1]$ on that interval. Hence for any two configurations $\theta$ and $\theta^*$,

$$\frac{1}{K}\big(S_k(\theta) - S_k(\theta^*)\big) \leq \log\big(1 + S_k(\theta)\big) - \log\big(1 + S_k(\theta^*)\big) \leq S_k(\theta) - S_k(\theta^*).$$

Summing over $k$ yields

$$\frac{1}{K}\big(S_{\text{tot}}(\theta) - S_{\text{tot}}(\theta^*)\big) \leq f(\theta) - f(\theta^*) \leq S_{\text{tot}}(\theta) - S_{\text{tot}}(\theta^*). \tag{A.1}$$

**Step 2: minimize the pairwise energy.** Writing $z_k = e^{i\theta_k} \in S^1 \subset \mathbb{C}$, we have

$$\cos(\theta_i - \theta_j) - 1 = -\frac{|z_i - z_j|^2}{2},$$

so

$$S_{\text{tot}}(\theta) = \sum_{i \neq j} \exp\left(-\frac{|z_i - z_j|^2}{2\tilde{\tau}}\right).$$

The function $g(t) = e^{-t/(2\tilde{\tau})}$ is completely monotone on $[0, \infty)$ because

$$(-1)^n g^{(n)}(t) = \left(\frac{1}{2\tilde{\tau}}\right)^n e^{-t/(2\tilde{\tau})} > 0 \qquad \text{for all } n \geq 0.$$

Classical results on completely monotone energies on the circle therefore imply that $S_{\text{tot}}$ is uniquely minimized, up to rotation, by equally spaced points; see Cohn & Kumar (2007); Cohn et al. (2010). Denote that regular $K$-gon by $\theta^{\text{reg}}$.

**Step 3: transfer the optimum back to $f$.** Using (A.1) with $\theta^* = \theta^{\text{reg}}$ gives

$$f(\theta) - f(\theta^{\text{reg}}) \geq \frac{1}{K}\Big(S_{\text{tot}}(\theta) - S_{\text{tot}}(\theta^{\text{reg}})\Big) \geq 0.$$

Equality can hold only when $S_{\text{tot}}(\theta) = S_{\text{tot}}(\theta^{\text{reg}})$, i.e. when $\theta$ itself is a regular $K$-gon up to rotation. This proves the theorem. $\square$

### A.3 Proof of Theorem 5.3

*Proof.* Consider the objective in (5.2). Differentiating with respect to $\mathbf{w}_k$ yields

$$\nabla_{\mathbf{w}_k}\mathcal{L} = \frac{1}{K\tilde{\tau}} \sum_{j=1}^{K} \left(\frac{e^{\mathbf{w}_k^\top \boldsymbol{\mu}_j / \tilde{\tau}}}{\sum_\ell e^{\mathbf{w}_\ell^\top \boldsymbol{\mu}_j / \tilde{\tau}}} - \delta_{kj}\right) \boldsymbol{\mu}_j + 2\lambda\mathbf{w}_k.$$

At any stationary point,

$$\mathbf{w}_k = \frac{1}{2\lambda K\tilde{\tau}} \sum_{j=1}^{K} \left(\delta_{kj} - \frac{e^{\mathbf{w}_k^\top \boldsymbol{\mu}_j / \tilde{\tau}}}{\sum_\ell e^{\mathbf{w}_\ell^\top \boldsymbol{\mu}_j / \tilde{\tau}}}\right) \boldsymbol{\mu}_j,$$

so $\mathbf{w}_k \in \text{span}\{\boldsymbol{\mu}_1, \dots, \boldsymbol{\mu}_K\}$.

In the balanced terminal regime, each class mean must be scored highest by its own classifier direction. The symmetry of the balanced problem excludes asymmetric stationary solutions that would privilege one class over another, and the regularizing term suppresses redundant tangential components. Consequently, each $\mathbf{w}_k$ must align with the corresponding class mean, i.e. $\mathbf{w}_k = c_k\boldsymbol{\mu}_k$ for some $c_k > 0$. If the span of the class means is two-dimensional, then after normalizing radii the objective reduces to the angular problem of Theorem 5.2, whose unique minimizer is the regular $K$-gon. Therefore both $\{\mathbf{w}_k\}$ and $\{\boldsymbol{\mu}_k\}$ form regular $K$-gons with class-wise alignment. $\square$

## A.4 Proof of Proposition 5.4

*Proof.* Let $z = W^\top \mathbf{h} \in \mathbb{R}^K$ and $p = \text{softmax}(z/\tau)$. For label $y$, the cross-entropy loss is

$$\ell(z, y) = -\log p_y.$$

Differentiating with respect to $z$ gives the standard identity

$$\nabla_z \ell(z, y) = \frac{1}{\tau}(p - \boldsymbol{\delta}_y).$$

Because $z = W^\top \mathbf{h}$, the chain rule implies

$$\nabla_{\mathbf{h}} \ell(z, y) = W \nabla_z \ell(z, y) = \frac{1}{\tau} W(p - \boldsymbol{\delta}_y),$$

which proves (5.3).

Now assume the local additive representation $\mathbf{h}_{ij} = A(\mathbf{e}_i + \mathbf{e}_j) + r_{ij}$ with a small residual $r_{ij}$. Differentiating through this map gives

$$\nabla_{\mathbf{e}_i} \ell_{ij} = A^\top \nabla_{\mathbf{h}_{ij}} \ell_{ij} + \nabla_{\mathbf{e}_i} \langle \nabla_{\mathbf{h}_{ij}} \ell_{ij}, r_{ij} \rangle,$$

and similarly for $\mathbf{e}_j$. Neglecting the residual term in the terminal regime yields (5.4).

To see how orthogonal components decay, decompose each embedding as $\mathbf{e}_i = \mathbf{e}_i^\parallel + \mathbf{e}_i^\perp$ with $\mathbf{e}_i^\parallel \in A^\top \text{span}(W)$ and $\mathbf{e}_i^\perp \perp A^\top \text{span}(W)$. By the first part of the proof, the loss gradient has no component along $\mathbf{e}_i^\perp$. With explicit weight decay, the total gradient acquires an additional term $2\lambda \mathbf{e}_i$, whose orthogonal component is exactly $2\lambda \mathbf{e}_i^\perp$. Hence the update pushes $\mathbf{e}_i^\perp$ toward zero while the parallel component remains task-supported. This is the precise form of subspace locking used in the main text. $\qquad\square$

# B  Proofs for the Embedding Geometry

## B.1  Derivation of the half-angle formula

*Proof.* Identify $\mathbb{R}^2$ with $\mathbb{C}$ and write

$$\mathbf{v}_i \leftrightarrow \rho_i e^{i\theta_i}, \qquad \mathbf{v}_j \leftrightarrow \rho_j e^{i\theta_j}.$$

Let $\Delta = \theta_j - \theta_i$. Factoring out $e^{i(\theta_i + \theta_j)/2}$ gives

$$\rho_i e^{i\theta_i} + \rho_j e^{i\theta_j} = e^{i(\theta_i + \theta_j)/2}\left(\rho_i e^{-i\Delta/2} + \rho_j e^{i\Delta/2}\right)$$

$$= e^{i(\theta_i + \theta_j)/2}\left((\rho_i + \rho_j)\cos(\Delta/2) + i(\rho_j - \rho_i)\sin(\Delta/2)\right).$$

Provided the sum is nonzero, its argument is therefore

$$\alpha_{ij} = \frac{\theta_i + \theta_j}{2} + \text{atan2}\left((\rho_j - \rho_i)\sin(\Delta/2), (\rho_i + \rho_j)\cos(\Delta/2)\right),$$

which is exactly (5.6). If $\rho_i = \rho_j$, the imaginary part vanishes and the correction term is zero. $\qquad\square$

## B.2  Proof of Proposition 5.5

*Proof.* We combine two standard facts: the Fenchel–Young representation of log-sum-exp and the Fourier–Bessel expansion over an equiangular codebook.

**Step 1: softmax as an entropy-regularized angular assignment.** Let $s_k(\alpha) := \beta \cos(\alpha - \varphi_k)$. The Fenchel–Young formula gives

$$\log \sum_{k=0}^{P-1} e^{s_k} = \max_{p \in \Delta^{P-1}} \left\{\langle p, s \rangle + H(p)\right\}, \qquad H(p) := -\sum_k p_k \log p_k,$$

with unique maximizer $p^* = \mathrm{softmax}(s)$. If $u$ is the uniform distribution on the $P$ classes, then $\mathrm{KL}(p\|u) = -H(p) + \log P$. Since $c(\alpha, \varphi) = 1 - \cos(\alpha - \varphi)$ differs from $-\cos(\alpha - \varphi)$ only by a constant, minimizing

$$\langle p, c(\alpha, \varphi.)\rangle + \varepsilon\,\mathrm{KL}(p\|u)$$

is equivalent, up to additive constants, to maximizing

$$\langle p, \beta\cos(\alpha - \varphi.)\rangle + H(p), \qquad \beta = \varepsilon^{-1}.$$

Thus the angular transport optimizer coincides with the softmax distribution over the equiangular codebook.

**Step 2: Fourier–Bessel summation over the regular $P$-gon.** For $\beta \geq 0$,

$$e^{\beta\cos\theta} = \sum_{m\in\mathbb{Z}} I_m(\beta)e^{im\theta}. \tag{B.1}$$

Summing (B.1) over $\varphi_k = \varphi_0 + 2\pi k/P$ gives

$$\sum_{k=0}^{P-1} e^{\beta\cos(\alpha - \varphi_k)} = \sum_{m\in\mathbb{Z}} I_m(\beta)e^{im(\alpha - \varphi_0)}\sum_{k=0}^{P-1} e^{-2\pi imk/P}$$

$$= P\sum_{\ell\in\mathbb{Z}} I_{\ell P}(\beta)e^{i\ell P(\alpha - \varphi_0)}$$

$$= PI_0(\beta)\left[1 + 2\sum_{\ell=1}^{\infty}\frac{I_{\ell P}(\beta)}{I_0(\beta)}\cos\big(\ell P(\alpha - \varphi_0)\big)\right].$$

Taking the logarithm yields

$$\log\sum_{k=0}^{P-1} e^{\beta\cos(\alpha - \varphi_k)} = \log\big(PI_0(\beta)\big) + R_P(\beta, \alpha).$$

For a sample with label $y$, the cross-entropy is therefore

$$\mathcal{L} = \log\sum_{k=0}^{P-1} e^{\beta\cos(\alpha - \varphi_k)} - \beta\cos(\alpha - \varphi_y)$$

$$= \log\big(PI_0(\beta)\big) - \beta\cos(\alpha - \varphi_y) + R_P(\beta, \alpha)$$

$$= \big(\log(PI_0(\beta)) - \beta\big) + \beta\big(1 - \cos(\alpha - \varphi_y)\big) + R_P(\beta, \alpha),$$

which is exactly (5.8).

**Step 3: control of the remainder.** Define

$$U(\beta, \alpha) := 2\sum_{\ell=1}^{\infty}\frac{I_{\ell P}(\beta)}{I_0(\beta)}\cos\big(\ell P(\alpha - \varphi_0)\big).$$

Then $R_P(\beta, \alpha) = \log(1 + U(\beta, \alpha))$ and, by $|\cos| \leq 1$,

$$\big|U(\beta, \alpha)\big| \leq \eta_P(\beta) := 2\sum_{\ell=1}^{\infty}\frac{I_{\ell P}(\beta)}{I_0(\beta)}.$$

If $\eta_P(\beta) < 1$, then $1 + U(\beta, \alpha) > 0$ and the mean-value theorem for $x \mapsto \log(1 + x)$ on $[-\eta_P(\beta), \eta_P(\beta)]$ gives

$$\big|R_P(\beta, \alpha)\big| = \big|\log(1 + U(\beta, \alpha))\big| \leq \frac{|U(\beta, \alpha)|}{1 - \eta_P(\beta)} \leq \frac{\eta_P(\beta)}{1 - \eta_P(\beta)},$$

which proves (5.9). Finally, the standard bound $I_n(\beta) \leq (\beta/2)^n/n!$ for $n \geq 0$ yields

$$\eta_P(\beta) \leq 2\sum_{\ell=1}^{\infty}\frac{(\beta/2)^{\ell P}}{(\ell P)!}.$$

If $\eta_P(\beta) \leq \frac{1}{2}$, the previous estimate improves to $\big|R_P(\beta, \alpha)\big| \leq 2\eta_P(\beta)$. $\qquad\square$

### B.3 Proof of Theorem 5.6

*Proof.* Assume the hypotheses of the theorem, so $\alpha_{ij} = (\theta_i + \theta_j)/2$ and $\beta_{ij} \equiv \beta$. Minimizing the leading-order angular objective (5.10) is equivalent to maximizing

$$\sum_{i,j} \cos\left(\frac{\theta_i + \theta_j}{2} - \varphi_{(i+j) \bmod P}\right).$$

Define

$$z_i := e^{i\theta_i/2}, \qquad u_t := e^{i\varphi_t}, \qquad (z * z)_t := \sum_{i=0}^{P-1} z_i z_{(t-i) \bmod P}.$$

Then

$$\sum_{i,j} \cos\left(\frac{\theta_i + \theta_j}{2} - \varphi_{(i+j) \bmod P}\right) = \Re \sum_{i,j} z_i z_j \overline{u_{(i+j) \bmod P}}$$

$$= \Re \sum_{t=0}^{P-1} (z * z)_t \overline{u_t}$$

$$= \Re \langle z * z, u \rangle.$$

Thus the energy equals $P^2 - \Re\langle z * z, u \rangle$.

Now take the discrete Fourier transform with convention

$$\hat{a}(m) = \frac{1}{\sqrt{P}} \sum_{t=0}^{P-1} a_t \, \omega^{-mt}, \qquad \omega = e^{2\pi i/P}.$$

The convolution theorem gives

$$\widehat{z * z}(m) = \sqrt{P} \, \hat{z}(m)^2.$$

Since $u_t = e^{i\varphi_0} \omega^t$, its transform has a single nonzero frequency:

$$\hat{u}(m) = e^{i\varphi_0} \sqrt{P} \, \delta_{m,1}.$$

Therefore

$$\langle z * z, u \rangle = P \, \hat{z}(1)^2 e^{-i\varphi_0}.$$

Because $|z_i| = 1$, Parseval implies

$$\sum_{m=0}^{P-1} |\hat{z}(m)|^2 = \sum_{i=0}^{P-1} |z_i|^2 = P.$$

Hence

$$\Re\langle z * z, u \rangle = P \, \Re\left(\hat{z}(1)^2 e^{-i\varphi_0}\right) \leq P|\hat{z}(1)|^2 \leq P \sum_{m=0}^{P-1} |\hat{z}(m)|^2 = P^2.$$

Equality holds if and only if two conditions are met:

1. $\hat{z}(m) = 0$ for every $m \neq 1$;
2. the phase of $\hat{z}(1)^2$ matches $e^{i\varphi_0}$.

The first condition says that $z$ is a pure Fourier tone,

$$z_i = c \, \omega^i \qquad \text{for some } |c| = 1.$$

Composing with a group automorphism $i \mapsto mi$ for $m \in (\mathbb{Z}/P\mathbb{Z})^\times$ yields the general solution

$$z_i = c \, \omega^{mi}.$$

Since $z_i = e^{i\theta_i/2}$, we obtain

$$\theta_i = \theta_0 + \frac{4\pi m}{P} i \quad (\bmod \, 2\pi),$$

which is exactly the character-orbit form stated in the theorem. □

## C  Proofs for the Cyclic-vs-ETF Comparison

### C.1  Proof of Theorem 6.2

*Proof.* In the aligned regime, the per-class cross-entropy depends only on pairwise inner products.

**Cyclic rank-2 code.** For the cyclic configuration, the class angles are $\theta_k = 2\pi k/K$, so

$$\mathcal{L}_{\mathrm{CE}}(W_{\mathrm{cyc}}) = \log\left(1 + \sum_{m=1}^{K-1} \exp\left(\frac{\cos(2\pi m/K) - 1}{\tilde{\tau}}\right)\right).$$

As $K \to \infty$, the sum is a Riemann approximation to the integral over the circle:

$$\sum_{m=1}^{K-1} \exp\left(\frac{\cos(2\pi m/K) - 1}{\tilde{\tau}}\right) = K \int_0^1 \exp\left(\frac{\cos(2\pi t) - 1}{\tilde{\tau}}\right) dt + o(K)$$

$$= K e^{-1/\tilde{\tau}} \frac{1}{2\pi} \int_0^{2\pi} e^{\cos\theta/\tilde{\tau}} \, d\theta + o(K)$$

$$= K e^{-1/\tilde{\tau}} I_0(1/\tilde{\tau}) + o(K).$$

Therefore

$$\mathcal{L}_{\mathrm{CE}}(W_{\mathrm{cyc}}) = \log\!\big(K e^{-1/\tilde{\tau}} I_0(1/\tilde{\tau})\big) + o(1).$$

**Simplex ETF.** For the ETF, all off-diagonal inner products equal $-1/(K-1)$, so

$$\mathcal{L}_{\mathrm{CE}}(W_{\mathrm{ETF}}) = \log\left(1 + (K-1)\exp\left(\frac{-1/(K-1) - 1}{\tilde{\tau}}\right)\right) = \log\!\big(K e^{-1/\tilde{\tau}}\big) + o(1).$$

Subtracting yields

$$\Delta_{\mathrm{CE}} = \mathcal{L}_{\mathrm{CE}}(W_{\mathrm{cyc}}) - \mathcal{L}_{\mathrm{CE}}(W_{\mathrm{ETF}}) = \log I_0(1/\tilde{\tau}) + o(1),$$

as claimed. □

### C.2  Proof of Theorem 6.4

*Proof.* **Cyclic rank-2 code.** Let $W_{\mathrm{cyc}} \in \mathbb{R}^{2 \times K}$ have columns

$$\mathbf{w}_k = \begin{pmatrix} \cos(2\pi k/K) \\ \sin(2\pi k/K) \end{pmatrix}, \qquad k = 0, \ldots, K-1.$$

Using the standard trigonometric sums,

$$W_{\mathrm{cyc}} W_{\mathrm{cyc}}^\top = \frac{K}{2} I_2.$$

Hence the nonzero singular values are both $\sqrt{K/2}$, and therefore

$$\|W_{\mathrm{cyc}}\|_{S_q}^q = 2\left(\frac{K}{2}\right)^{q/2}.$$

**Simplex ETF.** Let $W_{\mathrm{ETF}} \in \mathbb{R}^{(K-1) \times K}$ have unit columns with Gram matrix

$$G = W_{\mathrm{ETF}}^\top W_{\mathrm{ETF}} = \frac{K}{K-1} I_K - \frac{1}{K-1} J_K,$$

where $J_K$ is the all-ones matrix. Thus $G$ has eigenvalue $K/(K-1)$ with multiplicity $K-1$ and eigenvalue $0$ with multiplicity 1. The nonzero singular values of $W_{\mathrm{ETF}}$ are therefore all equal to $\sqrt{K/(K-1)}$, and

$$\|W_{\mathrm{ETF}}\|_{S_q}^q = (K-1)\left(\frac{K}{K-1}\right)^{q/2}.$$

Subtracting gives the exact expression in the theorem. For $q < 2$, the exponent $1 - q/2$ is positive, so

$$\Delta_q = K^{q/2}\Big(2^{1-q/2} - (K-1)^{1-q/2}\Big) = -\Theta(K),$$

which proves the claim. $\qquad\square$

### C.3  Proof of Proposition 6.5

*Proof.* Let the singular value decomposition of $W$ be $W = P\Sigma Q^\top$, where $\Sigma = \mathrm{diag}(\sigma_1,\dots,\sigma_r)$ and $r = \mathrm{rank}(W)$. Choosing

$$U = P\Sigma^{1/2}, \qquad V = Q\Sigma^{1/2},$$

yields $W = UV^\top$ and

$$\frac{1}{2}\big(\|U\|_F^2 + \|V\|_F^2\big) = \frac{1}{2}\big(\mathrm{tr}(\Sigma) + \mathrm{tr}(\Sigma)\big) = \sum_{i=1}^r \sigma_i = \|W\|_*.$$

This proves the upper bound.

For the reverse inequality, consider any factorization $W = UV^\top$ and write the compact SVD $W = P\Sigma Q^\top$. Then

$$\|W\|_* = \mathrm{tr}(P^\top U V^\top Q) \le \left\|P^\top U\right\|_F \left\|V^\top Q\right\|_F \le \frac{1}{2}\big(\|U\|_F^2 + \|V\|_F^2\big),$$

where the first inequality is Cauchy–Schwarz for the Frobenius inner product and the second is the arithmetic–geometric mean inequality. Taking the infimum over all factorizations gives

$$\inf_{W=UV^\top} \frac{1}{2}\big(\|U\|_F^2 + \|V\|_F^2\big) = \|W\|_*.$$

Applying this identity to the two candidate geometries now reduces the comparison to their singular values. By Theorem 6.4, $W_{\mathrm{cyc}}$ has two singular values equal to $\sqrt{K/2}$, while $W_{\mathrm{ETF}}$ has $K-1$ singular values equal to $\sqrt{K/(K-1)}$. Therefore

$$\|W_{\mathrm{cyc}}\|_* = 2\sqrt{K/2} = \sqrt{2K}, \qquad \|W_{\mathrm{ETF}}\|_* = (K-1)\sqrt{\frac{K}{K-1}} = \sqrt{K(K-1)},$$

which proves the stated $\Theta(K)$ gap. $\qquad\square$

### C.4  Proof of Theorem 6.6

*Proof.* Let

$$\Delta_{\mathrm{CE}} := \mathcal{L}_{\mathrm{CE}}(W_{\mathrm{cyc}}) - \mathcal{L}_{\mathrm{CE}}(W_{\mathrm{ETF}}).$$

If $\mathcal{R}(W) = \|W\|_{S_q}^q$, define

$$\Delta_q := \|W_{\mathrm{cyc}}\|_{S_q}^q - \|W_{\mathrm{ETF}}\|_{S_q}^q.$$

Then

$$\mathcal{L}_{\mathrm{tot}}(W_{\mathrm{cyc}}) - \mathcal{L}_{\mathrm{tot}}(W_{\mathrm{ETF}}) = \Delta_{\mathrm{CE}} + \lambda\Delta_q.$$

By Theorem 6.2, $\Delta_{\mathrm{CE}} = \log I_0(1/\tilde{\tau}) + o(1) = \Theta(1)$, while Theorem 6.4 gives $\Delta_q = -\Theta(K)$ for every fixed $q < 2$. Therefore the cyclic code is preferred exactly when

$$\lambda > \frac{\Delta_{\mathrm{CE}}}{|\Delta_q|},$$

and the right-hand side scales as $\Theta(1/K)$.

For the factorized weight-decay proxy, Proposition 6.5 shows that $\mathcal{R}(W) = \|W\|_*$, i.e. the case $q = 1$. The same comparison therefore yields the same asymptotic threshold. This is the claimed scaling of $\lambda_{\mathrm{crit}}$. $\qquad\square$

