# OpenReview forum: "Beyond Neural Collapse: Task-Intrinsic Geometry Governs Neural Representations in Modular Arithmetic"
_TMLR — Under review for TMLR_

### Review · Reviewer_xwB1 · 2026-07-09

**Summary Of Contributions:**

This paper studies why neural networks trained on modular addition learn a 2D circular representation instead of the simplex ETF structure predicted by Neural Collapse. The authors propose that the classifier weights first form a rank-2 cyclic structure, which then guides embeddings into the same subspace. They also analyze the trade-off between classification loss and regularization to explain why the cyclic solution can be preferred. the paper provides a nice perspective connecting task structure, regularization, and representation geometry. However, some theoretical claims rely on strong assumptions that are not fully validated by experiments.

**Audience:**

Yes

**Audience Explanation:**

The paper studies an important question in deep learning theory: how neural networks discover structured representations.

**Broader Impact Concerns:**

No major ethical concerns.

**Claims And Evidence:**

No

**Claims Explanation:**

the theory assumes several important properties, such as rank-2 classifier geometry, feature-classifier alignment, and the additive feature model. These assumptions are close to the final observation itself, so the paper does not fully explain why training dynamics lead to this regime.

The paper also needs more experiments showing the claimed "classifier first, embeddings follow" process, rather than only analyzing the final representation.

**Requested Changes:**

1. Add training-dynamics analysis to verify the "classifier first, embeddings follow" claim. For example, show how classifier rank, embedding rank, and alignment change during training.
2. Could clarify which parts are theoretical assumptions and which parts are empirically observed. Several results depend on assumptions such as rank-2 structure and homogeneous embedding norms.
3. May addd more experiments across different architectures, optimizers, weight decay settings, and modulus sizes to show whether the findings are general.
4. Provide a more detailed comparison with Neural Collapse and clarify whether this is a special property of modular arithmetic or a more general phenomenon.
5. Include more quantitative measurements of the learned geometry, such as effective rank and alignment metrics.

---

### Review · Reviewer_2Eiu · 2026-07-11

**Summary Of Contributions:**

This paper studies the feature geometry of the modular addition problem. The main goal of the paper is to provide theoretical explanation of why the last-layer features form a circular structure on a low-dimensional sphere rather than a high-dimensional ETF as in neural collapse. Specifically, the main contributions are summarized as follows:

- Assume that the features are in the 2-dimensional space, and the classifier's weights align with the feature, the circular structure is the global optimum. (Section 5.1)
- Assume the classifier's weight is in the 2-dim space, and under the feature additive assumptions, the gradient of the embedding will lines in this space, and the weight decay will remove the orthogonal component. (Section 5.2)
- Assume the feature lines in the 2-dim subspace, then the global optimal features is the equi-angular structure on the circle. (Section 5.3)
- The  equi-angular structure on the 2-dim circle has smaller error compare to ETF under Schatten norm or nuclear norm regularization. (Section 6)

**Strength**
- Providing theoretical explanations on the feature structure in modular addition problem is interesting.

**Weaknesses**

- The main results relays on many assumptions that are not properly justified:
    * Theorem 5.1 assume the features $h$ already align with the classifers' weights.
    * Proposition 5.4 assume additive feature model, i.e. $h_{i,j} = A(e_i + e_j),$ I don't see where this assumptions come from.

- The comparison between circular solutions and the ETF solutions are based on Schatten norm or nuclear norm, which is not justified . The authors provide a case where $W = UV$ and $L2$-regularization on $U,V$ become nuclear regularization on $W,$ but this is still a special case,  and it does not justify why there's bias towards solution with low  Schatten norm or nuclear norm.


- The paper is not well-written and is hard to follow. To name a few points:
    * In section 4, the authors mentioned a  two-layer ReLU network, but it is neither defined nor mentioned in later parts.
    * In section 5.2, the authors mentioned additive feature model, which is only defined and justified in the next section.
    * In section 4, the dimension of $e_k$ is 2, however, in section 5.3, $h = A e,$ with $A$ of size $2 \times d,$ which means $e$ is of dimension $d.$

**Audience:**

Yes

**Audience Explanation:**

The theoretical explanation of the feature structure in modular addition would be interesting to both theory and mechanistic interpretation community.

**Broader Impact Concerns:**

No broader impact concerns.

**Claims And Evidence:**

Yes

**Claims Explanation:**

I checked the proof, and they seem to be correct.

**Requested Changes:**

- Please define the problem setup properly in section, including the model, the dimension of each weights, and whether a parameter is fixed or trainable. It would be helpful to clarify each notation when it is defined.

- Please justify the assumptions mentioned in **Weaknesses** parts.

---

### Review · Reviewer_8nu7 · 2026-07-16

**Summary Of Contributions:**

This paper studies the emergence of low-dimensional cyclic representations in neural networks trained on modular arithmetic tasks. Specifically, the authors investigate why networks trained on modular addition learn two-dimensional circular representations rather than the high-dimensional simplex ETF geometry suggested by Neural Collapse (NC).

The paper proposes a theoretical explanation based on three stages: (1) the classifier weights first organize into a low-rank rank-2 cyclic geometry; (2) gradients from the classifier constrain subsequent embedding updates into the same low-dimensional subspace; and (3) the modular group structure induces Fourier-character solutions, resulting in equally spaced circular embeddings. The paper further compares this cyclic representation with the Neural Collapse simplex ETF solution and argues that low-rank regularization can make the cyclic representation preferable because it sacrifices only a bounded amount of cross-entropy performance while achieving a substantial reduction in complexity.

**Audience:**

Yes

**Audience Explanation:**

The topic is highly relevant to the TMLR audience. Understanding how neural networks develop structured internal representations is an important problem in deep learning theory and interpretability.

**Broader Impact Concerns:**

N/A.

**Claims And Evidence:**

No

**Claims Explanation:**

The paper presents several strong claims about the training dynamics and the mechanism underlying cyclic representations in modular arithmetic, but the provided evidence is insufficient to fully support these claims.



The main contribution of the paper is not the observation that modular addition produces circular representations, which has already been reported in previous studies, but rather the proposed explanation of *why* this happens. In particular, the paper argues that the classifier weights first collapse into a rank-2 geometry and that this classifier structure subsequently constrains embedding evolution. However, the paper does not provide direct experimental evidence for this temporal ordering.



The experiments mainly show the final learned geometry, such as PCA visualizations of embeddings and output weights. While these results demonstrate the existence of circular representations, they do not establish the claimed causal relationship between classifier formation and embedding alignment.



Several theoretical results also rely on assumptions whose validity in actual neural networks is not demonstrated. For example, the theoretical analysis assumes terminal feature-classifier alignment and an additive feature model of the form $h_{ij}\approx A(e_i+e_j)$. These assumptions may be reasonable approximations in the terminal phase, but the paper does not quantitatively verify that trained networks satisfy them.



In addition, the paper argues that weight decay induces a transition favoring cyclic rank-2 solutions and derives a scaling law $\lambda_{crit}=\Theta(1/K)$. However, no systematic experiments are provided to validate this prediction across different values of (K) or different regularization strengths.



Overall, the theoretical arguments are interesting, but the evidence currently does not convincingly support the broader mechanistic claims made in the paper.

**Requested Changes:**

**1. Add direct experiments validating the proposed classifier-first mechanism.**

The central claim of the paper is that classifier geometry emerges before embedding geometry. The authors should provide training trajectory analyses, including:



* effective rank of classifier weights over training;

* effective rank of embeddings over training;

* alignment between classifier and embedding subspaces.



---


**2. Add systematic weight decay experiments.**



The paper argues that low-rank regularization drives the preference for cyclic representations. The authors should evaluate different weight decay strengths and report.

---



**3. Validate the predicted scaling law.**



The paper derives:



$
\lambda_{crit}=\Theta(1/K)
$



but only provides a single numerical example.



Experiments across different modulus sizes (K) are needed to verify whether the predicted scaling behavior is observed empirically.



---



**4. Quantify the validity of theoretical assumptions.**



The paper should measure whether the assumptions used in the theory hold in trained networks. For example, additive feature decomposition and low-dimensional subspace confinement.



---

**5. Provide more complete experimental details.**



The paper should clearly specify: network architecture, hidden dimensions, optimizer, learning rate, training duration, random seeds, regularization settings, etc.



---



**6. Clarify the relationship with previous modular arithmetic literature.**



The paper should more clearly distinguish:



* previously known observations (circular representations, Fourier structure, grokking);

* new theoretical contributions introduced in this work.



---



**7. Moderate claims regarding Neural Collapse.**



The comparison with Neural Collapse should be framed more carefully. The results appear to show that modular arithmetic represents a structured task regime where classical NC assumptions may not apply, rather than demonstrating that Neural Collapse is generally inadequate.